# Sociogenetic Organization of the Red Honey Ant (*Melophorus bagoti*)

**DOI:** 10.3390/insects11110755

**Published:** 2020-11-04

**Authors:** Nathan Lecocq de Pletincx, Serge Aron

**Affiliations:** Evolutionary Biology & Ecology, Université Libre de Bruxelles, B-1050 Brussels, Belgium; saron@ulb.ac.be

**Keywords:** population structure, polyandry, worker reproduction, worker subcaste determination, *Melophorus*

## Abstract

**Simple Summary:**

Monogamy is thought to be a major factor having favored the evolution of a non- reproductive worker caste in eusocial insects because it optimizes the relatedness among colony members. However, polyandry evolved secondarily in a large number of species. By increasing the genetic diversity within colonies, multiple mating can enhance worker task efficiency and resistance to diseases. Polyandry may also favor social harmony by reducing worker–queen conflict over male parentage. This is because in colonies headed by a single, multiple-mated queen, workers can increase their inclusive fitness by rearing their brothers (queen sons) rather than their nephews (offspring of other workers). Using DNA microsatellites, we showed that nests of the red honey ant, *Melophorus bagoti*, are headed by a single, multiple-mated queen. Morphometric analyses revealed two distinct worker subcastes: majors and minors; yet, we found no relationship between worker patriline and worker subcaste. Workers can produce males in the presence of the queen under natural conditions, which contrasts with predictions of inclusive fitness theory.

**Abstract:**

Kin selection and inclusive fitness are thought to be key factors explaining the reproductive altruism displayed by workers in eusocial insect species. However, when a colony’s queen has mated with <2 males, workers may increase their fitness by producing their own male offspring. Conversely, when the queen has mated with ≥2 males, workers are expected to increase their inclusive fitness by eschewing the production of their sons and preventing other workers from reproducing as well. Here, we investigated sociogenetic structure and worker reproduction in the red honey ant, *Melophorus bagoti*. Morphometric analyses revealed that workers belong to one of two distinct subcastes: they are either majors or minors. Using DNA microsatellite markers, we showed that all the colonies had a single, multiple-mated queen and that there was no relationship between worker patriline and worker subcaste. Furthermore, we found that workers were producing males in the presence of the queen, which contrasts with the predictions of inclusive fitness theory. Although our results are based on a small sample, they can serve as the foundation for future research examining worker reproduction in *M. bagoti*.

## 1. Introduction

Reproductive altruism, whereby individuals forgo reproducing themselves to help others produce offspring, has generally been explained by evoking Hamilton’s model of kin selection and inclusive fitness theory [1,2]. Relatedness is critical for the evolution of reproductive altruism: it can enhance the helpers’ indirect fitness, since copies of the helpers’ genes are passed on to the subsequent generation by the beneficiary of the altruistic act. Thus, selection will favor altruism if altruists gain more in indirect fitness than they lose by not reproducing. Lifetime monogamy is a major prerequisite in the evolution of reproductive altruism because group members are thus equally related to their siblings and their own offspring (*r* = 0.5) [3]. Consequently, any infinitesimal benefit of helping one’s parents produce new offspring should favor the evolution of a non-reproductive helper caste. Ultimately, lifetime monogamy may be evolutionarily sufficient to select for permanently sterile worker castes, such as those seen in eusocial species [3,4,5].

Complementary support for the monogamy hypothesis comes from evidence that the ancestral reproductive system in eusocial insects (i.e., ants, termites, social bees, and wasps) was likely characterized by monogyny (i.e., the presence of one queen per colony) and monandry (i.e., queens have a single mate) [4,6,7]. However, multiple mating has secondarily evolved numerous times in ants, bees, and wasps [7]. Polyandry provides multiple fitness benefits [8]. For example, it enhances colony-level fitness by increasing the genetic diversity of the work force, which results in a more efficient division of labor and enhanced foraging performance [9], greater resistance to pathogens [10,11,12], and better balanced colony homeostasis [13,14]. In some ants, polyandry has been found to influence worker size and morphology in such a way that workers are partitioned into distinct physical subcastes based on paternity [15,16,17,18,19]. It has been suggested that this genetically mediated mechanism could allow colonies to rapidly alter caste ratios as their needs change [15,20]. Finally, polyandry by queens is predicted to reduce worker–queen conflict over male parentage within colonies [21,22,23,24]. In many species of eusocial Hymenoptera, workers have retained their ovaries and can lay unfertilized haploid eggs that produce male offspring [25,26]. However, workers may not reproduce, either because they refrain from laying eggs (i.e., self-restraint) or because their eggs are culled by the queen or other workers (i.e., queen policing or worker policing, respectively). Kin selection theory predicts that, in a monandrous system, workers can maximize their inclusive fitness by rearing their sons (*r* = 0.5) and nephews (offspring of other workers, *r* = 0.375) rather than their brothers (*r* = 0.25). However, as queen mating frequency increases, the relatedness between workers and their nephews decreases. When the queen has more than two mates, workers will be more related to their brothers (*r* = 0.25) than to their nephews (*r* < 0.25), and worker reproduction should be hampered by self-restraint and policing. Indeed, past studies have found that worker reproduction is lower in species in which workers are more related to their brothers than to their nephews [27,28,29].

*Melophorus* is an exceptionally diverse ant genus whose distribution is restricted to the hot arid and semi-arid regions of Australia [30,31]. The genus has attracted a lot of attention because of its remarkable navigation abilities [32,33] and resistance to heat stress [34,35]. In contrast, the sociogenetic organization and reproductive strategies of *Melophorus* species remain largely unexplored. Here, we present the first study to examine population and colony genetic structure in this genus, using the red honey ant, *Melophorus bagoti*, as a biological model. This species displays considerable variation in worker body size [36,37]. Large workers develop into ‘honey pots’ and store liquid food in their gasters; small workers perform the daily maintenance tasks that are necessary for colony growth. Field research has suggested that *M. bagoti* queens have multiple mates and can independently found new colonies (i.e., without the help of workers; [37]).

First, we developed a genomic library of microsatellite markers and used 12 polymorphic loci to characterize the genetic structure of colonies, including the number of queens per colony and the queen mating frequency. Second, we analyzed worker morphometry. We discovered the existence of two distinct worker subcastes; we then investigated whether there was a relationship between worker patriline and worker subcaste. Third, to assess whether workers were reproducing under natural conditions, we determined the maternity of males collected from colonies in the field.

## 2. Materials and Methods

### 2.1. Field Collection and Sampling

*Melophorus bagoti* is the most thermophilic ant in Australia [34]. The species lives in the low-shrub and grassland deserts found in the country’s center and builds large colonies that extend deep into the hard soil [36]. Thirteen colonies were collected in January 2019 in an area located to the south of Alice Springs (Northern Territory); they came from two populations located 13 km apart: CAT (23°45′32.5″ S, 133°53′01.0″ E) and SPR (23°50′37.9″ S, 133°57′20.7″ E) (GPS coordinates of each colony are given in Appendix A). We sampled all mated queens, workers, and, if present, female and male reproductives in each colony. Three colonies only produced sexuals: one producing female, and two producing male reproductives. When reproductives were present, they were immediately stored in 98% ethanol for use in subsequent genetic analyses. The queens and workers were brought back to Belgium, where they were maintained under laboratory conditions (28 ± 2 °C; 12 h:12 h light/dark photoperiod) and fed sugar water and cockroaches.

### 2.2. Microsatellite Markers and Genotyping

To develop the genomic library, total genomic DNA was extracted from a pool of nine workers using the Qiagen DNeasy Blood and Tissue Kit (Qiagen Inc., Valencia, CA, USA) in accordance with the manufacturer’s instructions. Non-enriched genomic libraries were constructed [38,39], and the DNA was sequenced using Illumina MiSeq (GIGA Platform, Liège, Belgium). The resulting paired-end reads were aligned using PANDAseq v. 2.10 [40], yielding a total of 1,997,698 reads. Microsatellite motifs were identified using QDD v. 3.1 [41]. We obtained 37,273 (1.9%) sequences that contained at least one microsatellite motif. We selected 36 candidate loci based on the following criteria: (i) a minimum distance of 20 bp between the primer and the microsatellite motif; (ii) the presence of at least 9 microsatellite repeats; (iii) the existence of only pure microsatellites (i.e., a single motif within the fragment); (iv) a weak alignment score (i.e., weak complementarity) between the primers and the amplicon; (v) an expected PCR product size of 100–300 bp.

To perform the genotyping, DNA was extracted using the Chelex process (Bio-Rad^®^, Hercules, CA, USA; [42]). For the adults, two legs were used. Samples were ground for 1 min at 20 Hz in 100 µL of 5% Chelex and then incubated for 1 h 30 min at 85 °C. After 3 min of centrifugation at 12,000 rpm, 40 µL of the supernatant was placed in a 1.5-mL tube. The eggs were used in their entirety; extraction took place in 20 µL of 5% Chelex. Sperm DNA was extracted using the Chelex procedure as described in [43]. To confirm microsatellite amplification and test for the presence of polymorphism, a fluorochrome was assigned to each microsatellite marker using a three-primer PCR approach [44]. The 36 markers were then tested on seven *M. bagoti* workers (each from a different colony) using a standard simplex PCR employing MyTaq DNA polymerase (Bioline, London, UK) in accordance with the manufacturer’s instructions. A PCR volume of 12.5 µL and an annealing temperature of 58 °C were used. Finally, 14 polymorphic microsatellite loci were retained for the subsequent genetic analyses (Appendix A). We designed two four-marker multiplex assays and one five-marker multiplex assay using Multiplex Manager v. 1.0 ([45]; Appendix A). These multiplex PCRs were carried out using a QIAGEN Type-it Microsatellite PCR Kit (10-µL reactions as per the manufacturer’s instructions and a 58 °C annealing temperature) and a Thermal Duo-Cycler (VWR, Berntsen, Denmark). All genotypes are available in Appendix A.

### 2.3. Genetic Analyses

#### 2.3.1. Population Genetic Structure

We compared allele frequencies between the two populations at each locus with a *G*-test implemented via GENEPOP v. 4.7.0 [46]; we then used Fisher’s method to carry out an overall test across all the loci. Locus-specific allele frequencies differed between the populations (all *p* values = 0). However, the overall test yielded inconclusive results (*G*-test, Fisher’s method, *p* < 0.49). Consequently, to be conservative, the two populations were treated as genetically distinct in our analyses.

We tested for linkage disequilibrium (LD) and deviation from Hardy–Weinberg equilibrium (HWE) within each of the two populations using GENEPOP v. 4.7.0. Because the colonies each contained several patrilines (i.e., lineages of offspring produced by different fathers; see Results), the composite LD test [47] was conducted on the separate patrilines (number of patrilines across all the colonies sampled = 39) instead of on the colonies (*N* = 13). Because the multiple genotypes arising from a single matriline are not independent, a resampling procedure was used when testing for deviation from HWE [48]: a single individual was selected at random per colony, and this procedure was repeated 15 times (number of individuals per replicate = 13). Each replicate was analyzed using an exact test of HWE [47]. The 15 *p*-values were treated with a modified Lancaster procedure (a generalization of the Fisher’s method) to deal with dependence among replicates [49]. We controlled for genotyping errors due to null alleles and allele dropouts using MICRO-CHECKER [50], which was applied to the 15 replicates created for HWE testing.

Finally, we investigated the genetic structure of the two populations. We estimated the number of alleles per locus, allele frequencies, observed heterozygosity (H_O_), and expected heterozygosity (H_E_) using SPaGeDi v. 1.5.a [51] (Appendix A). Population genetic structure was examined using analysis of molecular variance (AMOVA). Hierarchical analysis of genetic variation and the calculation of *F*-statistics were performed at four levels using Arlequin v. 3.5.2.2 [52]: between populations (*F*_CT_), among colonies within populations (*F*_SC_), among individuals within colonies (*F*_IS_), and within individuals (*F*_IT_). The significance of the fixation indices was tested using non-parametric permutations (number of permutations = 1023). Isolation-by-distance patterns were investigated by plotting the coefficients [*F*_ST_/(1 − *F*_ST_)] for the pairs of colonies against the natural log-transformed geographical distances [53]. The significance of the regression line’s slope was estimated using 1000 permutations of the colony locations and SPaGeDi v. 1.5.a. Relatedness coefficients were estimated using SPaGeDi v. 1.5.a.

#### 2.3.2. Queen Number and Mating Frequency

The minimum number of queens per colony was determined based on the field data. Of the 13 colonies sampled, 6 had one queen; no queens were found in the 7 other colonies. Queen pedigree was either determined by genotyping the queens or by reconstructing the queen’s genotype based on the workers’ genotypes. This approach was straightforward because *M. bagoti* colonies were headed by a single queen (see Results). All genotypes reconstructions were confirmed with the maximum likelihood method implemented in the program COLONY v. 2.0.6.4 [54].

Queen mating frequency was inferred via three methods: (i) using the genotypes of the queens and their worker offspring produced under controlled laboratory conditions; (ii) using the genotype of the sperm stored in the queens’ spermathecae; (iii) using the genotypes of workers collected in the field. For the workers collected in the field, we determined the minimum number of males who made contributions to each colony’s progeny based on worker genotype (i.e., the absolute number of mates per queen [*k_a_*]). Because males may contribute unequally to offspring production, we also estimated the effective mating frequency (*k_e_*) using the method developed by [55] (Equation (1)):(1)$$k_{e}=\frac{{\left(n-1\right)}^{2}}{\left[\sum_{i=1}^{k}p_{i}^{2}\right]\left(n+1\right)\left(n-2\right)+3-n}$$
where *n* is the sample size and *p_i_* is the proportional contribution of the *i*th male to the brood. Note that, at low sample sizes, this estimator tends to overestimate *k_e_* values, which may result in *k_e_* being slightly higher than *k_a_*; however, this estimator performs better than other estimators proposed to date [55].

We calculated the probability of two males bearing the same allele at each locus using the Equation (2) [56]:(2)$$d=\underset{j}{\prod }\underset{i}{\sum }p_{ij}{}^{2}$$
where *p_ij_* is the population frequency of allele *i* at locus *j*.

#### 2.3.3. Worker Subcaste Identification and Characterization

To characterize worker size, we measured the head width (eyes included) of 157 workers (mean number of workers per colony ± SD = 39.25 ± 5.91, *N* = 4 colonies). To explore worker allometry (i.e., the proportionality of body parts), we measured the length of the hind tibia for 143 of the 157 workers (mean number of workers per colony ± SD = 35.75 ± 8.26), and we performed a regression between head width and tibia length (both natural log transformed). In such analyses, when the slope of the regression line is significantly different from 1, an allometric relationship exists [57]. To complete our characterization of morphological variation in *M. bagoti*, we applied the same methodology to 15 female reproductives. All the measurements were made to the nearest 0.01 mm using a MZ6 stereomicroscope (Leica Microsystems, Wetzlar, Germany).

Workers of *M. bagoti* were found to cluster into two morphologically distinct subcastes: the major subcaste and the minor subcaste (see Results). To test whether worker patriline affected worker subcaste, we performed a Fisher’s exact test on the data for each colony. We combined the information gathered from each colony and analyzed this dataset using the weighted Z-method [58] to obtain an overall *p*-value. The strength of the bias in patriline representation between the majors and minors was estimated by measuring the effect sizes using Cramer’s V (0 = no association, 1 = maximum association; [59]). The dataset contained information on 192 workers: the 157 workers mentioned above plus 35 additional workers that were categorized as majors or minors based on visual assessments and for which we had genotypes at all 12 loci (mean number of workers per colony ± SD = 48 ± 8.83). It was easy to visually distinguish between the two subcastes because the majors had much larger head widths than did the minors.

#### 2.3.4. Maternity of the Males

We determined whether the males collected in the field were queen produced or worker produced. We genotyped 38 males taken from two colonies (18 from colony CAT3 and 20 from colony CAT6). No males were found in any of the other colonies. The genotypes of the males were compared to those of the queens and workers. Because males are haploid, a queen’s sons must carry a queen-derived allele at all their loci and should not display more than two alleles at a single locus. A worker’s sons have an equal probability of having an allele derived from the worker’s mother or father at each locus. Thus, any males carrying a non-queen-derived allele are the sons of a worker. Because worker-produced males could have queen-derived alleles at all their loci solely by chance, the probability of detecting a worker-produced male was estimated using the Equation (3) [60]:(3)$$P=\overset{n}{\underset{1}{\sum }}p_{i}\times \left(1-0.5^{l_{i}}\right)$$
where *n* is the number of patrilines in the colony, *p_i_* is the proportional contribution of the *i*th father to the brood, and *l_i_* is the number of informative loci analyzed for the *i*th patriline. An informative locus is one where the queen and her mate(s) have different alleles such that the workers’ offspring may inherit an allele that the queen does not have.

To check whether majors and minors’ ovarian development differed, we dissected 20 majors and 20 minors from different orphaned colonies.

## 3. Results

### 3.1. Population Genetic Structure

We found a significant linkage disequilibrium for three pairs of microsatellite loci. Consequently, two loci were excluded from our analyses, leaving us with 12 polymorphic markers (Appendix A). Among the latter, three loci had null alleles, which were coded as missing data. This procedure did not affect our results. There was a significant departure from HWE for 2 of the 12 microsatellite markers in each population (CAT: Mb06 and Mb24; SPR: Mb21 and Mb28). Although these loci may not be neutrally behaving Mendelian markers, including them in our analyses did not bias our data. The number of alleles per marker ranged from 4 to 22 in the CAT population and from 5 to 22 in the SPR population. The mean observed multi-locus heterozygosity was H_O_ = 0.86 (range: 0.57–0.98) in the CAT population and H_O_ = 0.86 (range: 0.57–1) in the SPR population; the mean expected multi-locus heterozygosity in the two populations was H_E_ = 0.83 (range: 0.58–0.94) and H_E_ = 0.83 (range: 0.54–0.93), respectively.

The hierarchical analysis of genetic variation revealed that the two populations were weakly but significantly different (*F*_CT_ = 0.018; permutation test, *p* = 0.022). The colonies were genetically distinct, as revealed by their marked degree of genetic differentiation (*F*_SC_ = 0.22, *p* = 0). There was no isolation by distance among colonies at the population scale (CAT: slope coefficient ± SE_Jackknife_ = 0.006 ± 0.010; *p* = 0.75; *N* = 7 colonies; SPR: slope coefficient ± SE_Jackknife_ = 0.005 ± 0.028; *p* = 0.69; *N* = 6 colonies). The inbreeding coefficient *F*_IT_ was low and not significantly different from 0, a result that strongly suggested that mating was random within each population (CAT: *F*_IT =_ −0.003, *p* = 0.81; SPR: *F*_IT =_ −0.003, *p* = 0.84; global: *F*_IT_ = 0.005, *p* = 0.96). The hypothesis of random mating was also supported by the absence of genetic relatedness between the queens and their mates (*r_q-m_* ± SE_Jackknife_ = −0.012 ± 0.017; range: −0.23–0.20; Student *t* test for the difference from 0: *t* = −0.57, *df* = 38, *p* = 0.57) (Table 1).

### 3.2. Queen Number and Mating Frequency

In all 13 colonies, workers appeared to have been produced by a single queen based on the genotype data. The mean genetic relatedness between the workers and the queen for a given colony, *r_q-w_*, was 0.50 (SE_Jackkniffe_ = 0.01; *N* = 13 colonies) (Table 1), as expected under conditions of monogyny. Interestingly, at all her loci, the queen from colony CAT4 had an allele from one of the workers’ fathers (the same father at each locus) and an allele from the workers’ mother, which is evidence that this queen was the workers’ sister. Similarly, pedigree analyses indicated that the 10 female reproductives in colony CAT4 were the queen’s sisters.

The genotypes obtained using the three different methods—the mother-offspring genetic combinations obtained in the laboratory (mean number of offspring genotyped per queen ± SD = 8.33 ± 2.31, *N* = 3 colonies), sperm characterization (number of queens’ sperm = 6), and the workers collected in the field (*N* = 13 colonies)—yielded consistent results and indicated that the queens had multiple mates. Based on the genotypes of the workers collected in the field, the mean absolute and effective number of mates per queen were *k*_a_ = 3 (SD = 0.71; range: 2–4) and *k_e_* = 2.82 (SD = 0.67; range: 1.51–3.78; Equation (1)), respectively. These results are robust since the probability of two males bearing the same alleles at all loci was very low (*d* = 1.6 × 10^−11^; Equation (2)). The queens’ inferred mates were unrelated (*r_m-m_* ± SE_Jackknife_ = −0.006 ± 0.015; range: −0.21–0.53; Student *t* test for the difference from 0: *t* = −0.29, *df* = 41, *p* = 0.77). As expected in a monogynous and polyandrous system, the mean genetic relatedness among nestmate workers was *r_w-w_ =* 0.44 (SE_Jackkniffe_ = 0.01) (Table 1), which was significantly different from the value of 0.75 that would be expected in a monogynous and monandrous system (Student *t* test: *t* = −20.34, *df* = 12, *p* = 0).

### 3.3. Worker Subcaste Identification and Characterisation

Worker head width was bimodally distributed, with no overlap between the two groups (Figure 1a). The slope of the regression line describing the relationship between workers’ head width and tibia length was 2.07 (both variables were natural log transformed; linear regression: SE = 0.07, *F*_1,142_ = 809.3, *p* = 0) (Figure 1b), indicating the presence of allometry. However, when the analysis was performed within each group of workers, the relationship was isometric. The group-specific regression lines were parallel (relative to tibia length, one group had larger heads than did the other group), which is what would be expected in a case of complete, well-established dimorphism [57]. These results confirm that *M. bagoti* workers belong to two morphologically distinct subcastes: namely majors (i.e., with larger head widths) and minors (i.e., with smaller head widths). Majors and minors also differed in ovarian morphology; upon dissection, majors were found to have two ovarioles per ovary, while minors only had one. Female reproductives’ and workers’ head width distribution did not overlap (Figure 1a). The slope of the regression line describing the relationship between female reproductives’ head width and tibia length was significantly different from 1 (slope ± SE = 0.67 ± 0.13, *t* test, *t* = 5.20, *df* = 13, *p* = 0.023; Figure 1b), indicating allometry in the queen caste.

The results of the overall (*p* = 0.08) and colony-specific (Table 1) Fisher’s exact tests showed no significant effect of patriline on worker subcaste. Indeed, the effect sizes (Cramer’s V) ranged from 0.19 to 0.31 (Table 1; mean ± SD = 0.24 ± 0.06), which indicates a low level of association between the two variables.

### 3.4. Maternity of the Males

In the field, males were found in two colonies: CAT3 and CAT6. Both colonies were queenright. In CAT6, all 20 of the genotyped males only had queen-derived alleles, indicating that they were the queen’s sons. In colony CAT3, 6 of 18 genotyped males bore queen-derived alleles at all 12 loci. However, the 12 other males carried alleles from the queen’s mates at several loci, showing that they were the sons of workers. Workers from three of the four patrilines produced male offspring. The mean probability of detecting a male that had been produced by a worker in both CAT3 and CAT6 was *p* = 0.999 (Equation (3)), indicating that it was extremely unlikely that the son of a worker would be attributed to a queen.

We also genotyped 67 eggs taken from three queenless laboratory colonies. All the eggs were haploid, and their genotypes were consistent with them being produced by the colonies’ workers. In all three colonies, the eggs carried alleles derived from the workers’ different fathers, indicating that workers from different patrilines were reproducing. Ovary dissections revealed the presence of yellow bodies and oocytes in both majors and minors, providing evidence that workers in the two subcastes can activate their ovaries in orphaned colonies.

## 4. Discussion

We found that the colonies of the desert ant *Melophorus bagoti* were headed by single, multiple-mated queens and contained workers belonging to two morphologically distinct subcastes (majors and minors). Colonies were genetically differentiated and exhibited no signs of isolation by distance, suggesting that new colonies are independently founded. Our results also show that workers can reproduce under natural conditions in presence of the queen and that both major and minor workers can lay haploid eggs in queenless nests.

Our genetic results support previous field observations [37] that *M. bagoti* is polyandrous and uses independent colony foundation. In ants, mating behavior, colony foundation type, and local genetic variation are typically linked [61,62]. In species displaying male aggregation, males and virgin queens fly away from their natal colonies and participate in mating swarms. In species displaying female calling, virgin queens “call” males by releasing sex pheromones; they can either mate close to their natal colony or disperse before mating. Once mated, queens may (i) further disperse by flight or on foot to independently found a colony, (ii) re-enter the natal colony and disperse at a later time, potentially accompanied by part of the worker force (i.e., dependent colony foundation), or (iii) re-enter the natal colony and then reproduce (reviewed in [63,64]). In independent colony foundation, queens most often disperse by flight far from the natal nest, and thus no isolation by distance is seen at the population scale. In dependent colony foundation, queens disperse by foot, which often leads to the genetic structuration of populations. Interestingly, field observations suggest that *M. bagoti* queens may use alternative mating and dispersal strategies [37]. First, virgin queens and males have been seen to fly away from colonies in synchrony, consistent with the male aggregation syndrome. However, virgin queens have also been observed in situations where they are surrounded by several males, who are running around on the ground, which is indicative of the female calling syndrome. Second, queens have been reported to either (i) independently found new colonies far away from their natal nests or (ii) disperse on foot from their natal nests, which suggests that newly mated queens can re-enter their natal nests before dispersal. The results of our genetic analyses provide support for some of these observations. They indicate that the two study populations were not genetically structured, as would be expected for a species displaying the male aggregation syndrome. Furthermore, they show that queens can inherit their mother’s colony, which confirms that young queens can re-enter and reproduce within their natal nests. Taken together, these findings highlight that mating and dispersal behaviors are plastic in *M. bagoti*. They also emphasize the need to clarify under which conditions and to what extent queens use such alternative mating and dispersal strategies.

Our morphometric analyses showed that *M. bagoti* has two distinct worker subcastes: majors and minors. Two factors are thought to play a decisive role in the developmental determination of adult worker morphology in ants [65]: (i) the critical size at which a larva initiates metamorphosis and (ii) the growth rates at which different organs grow. When these two factors undergo reprogramming, new phenotypes are produced. The major and minor phenotypes in *M. bagoti* workers indicate that critical sizes and growth rates are different between the two subcastes [65]. Indeed, variation in worker size can be explained by changes in critical size, whereas the difference in the relationship between head width and tibia length in the two subcastes suggests that the growth rates of these two body parts also differ. Yet a third combination of critical sizes and growth rates are seemingly used to produce queens. Developmental rules can be reprogrammed in different ways, which fall along a continuum [66]. On one end, castes may arise from genomic differences [15,16,17,18,19]. On the other end, castes may be the result of the differential expression of genes within the same genome due to contrasted growth environments [67,68]. Here, we found no significant effect of worker patriline on worker subcaste in *M. bagoti*. That said, the overall *p*-value was marginally insignificant (0.08), and the effect sizes were pronounced, suggesting that worker subcaste is not strictly environmentally determined in this species. It seems possible that a combination of genomic and differential gene expression influences [69] have led to the reprogramming of developmental rules in *M. bagoti* and the emergence of three distinct female phenotypes: queen, major, and minor.

We found that colonies of *M. bagoti* are headed by a single queen who had mated with two to four males; mean effective mating frequency was 2.82. In monogynous systems, when the queen has multiple mates (effective paternity >2), the workers are more related to the queen’s sons than to the sons of other workers (see Introduction; [22]). This situation should result in self-restraint or policing and in the absence of any worker-produced males. Here, only two colonies in the field had produced males. Both were queenright, and one contained worker-produced males. In CAT3, where the queen’s effective mating frequency was 3.53, workers were the mothers of 12 of the 18 males. There may be several reasons why workers reproduce despite the presence of the queen. First, in species with extended nest structures, workers may be scattered throughout the colony and can thus escape the policing efforts of the queen and/or their fellow workers [70]. This explanation could clarify our results because *M. bagoti* has large nests composed of multiple rooms that hold just a few workers each. Second, a decline in queen fertility may trigger reproduction by workers [71,72]. Third, worker policing may not evolve if it decreases colony efficiency, even when queens have multiple mates [73,74]. To test this ‘colony productivity’ hypothesis in *M. bagoti*, we would need to determine whether policing of worker reproduction occurs and to quantify the associated changes in fitness, which has never yet been done in polyandrous eusocial insects [11]. Because of our very limited sample size, further studies are needed to confirm that worker reproduction is occurring in queenright *M. bagoti* colonies and to characterize its proximate and ultimate causes.

## 5. Conclusions

This study shows that colonies of the red honey ant, *M. bagoti*, are headed by a single queen with multiple mates and supports that newly mated queens may use different mating and dispersal strategies. It also shows that workers belong to two distinct morphological castes and that worker patriline does not significantly affect worker subcaste. Moreover, it suggests that both queens and workers can produce males in queenright colonies in nature. Future studies should, however, examine worker reproduction, worker policing, and polyandry in this species to gain insight into the diversity of mating strategies and sociogenetic organizations of eusocial insects.

## Figures and Tables

**Figure 1 insects-11-00755-f001:**
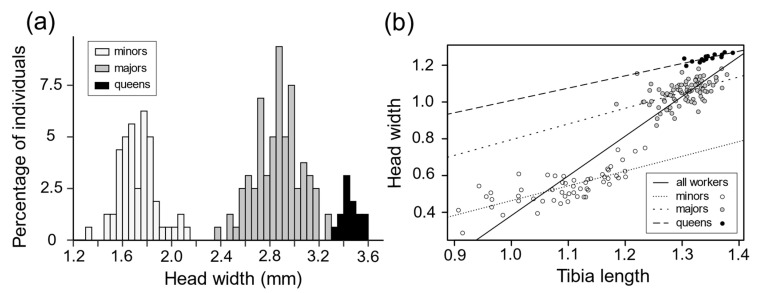
(**a**) Distribution of worker and female reproductive (queens) head width based on measurements from 157 workers (60 minors and 97 majors) originating from 4 colonies and from 15 female reproductive out of one colony. (**b**) Relationship between head width and tibia length based on measurements from 143 workers (55 minors and 88 majors) and 15 queens. The two variables were natural log-transformed. When the measurements for the two worker subcastes were combined (solid line), the regression analysis revealed the presence of allometry (linear regression: slope ± SE = 2.07 ± 0.07, *F*_1,142_ = 809.3, *p* = 0). In the worker subcaste-specific regressions, the slopes for the minors (line with short dashes) and the majors (line with long dashes) were not significantly different from 1, indicating that there was no further allometry within subcastes (minors: slope ± SE = 0.80 ± 0.10, *t* test, *t* = −1.94, *df* = 53, *p* = 0.057; majors: slope ± SE = 0.86 ± 0.16, *t* test, *t* = −0.89, *df* = 86, *p* = 0.38).

**Table 1 insects-11-00755-t001:** Mating system, colony and population genetic structure, and worker subcaste determination in *Melophorus bagoti.* The table indicates, for each colony, the number of workers genotyped to determine colony and population genetic structure; the absolute (*k_a_*) and effective (*k_e_*) mating frequency of the queen; relatedness among nestmate workers (*r_w-w_*), the queen and her mates (*r_q-m_*), and the queens’ mates (*r_m-m_*). Also given, for the subcaste-specific analyses, are the number of major and minor workers that were genotyped as well as the Fisher *p*-values and Cramer’s V values. The mean Fisher *p*-values were calculated by combining the data from the four colonies included in this analysis. The numbers of patrilines identified from the genotypes are in brackets.

Colony	Workers	*k_a_*	*k_e_*	*r_w-w_*	*r_q-m_*	*r_m-m_*	Majors	Minors	Fisher *p*-Value	Cramer’s V
CAT1	15	3	3.42	0.39	0.02	0.04				
CAT2	15	3	2.41	0.39	−0.04	−0.03	18 (3)	20 (3)	0.56	0.19
CAT3	15	4	3.53	0.40	−0.04	0.08	30 (4)	29 (4)	0.31	0.26
CAT4	15	2	1.51	0.55	0.02	0.00	30 (2)	20 (2)	0.11	0.20
CAT5	15	2	2.12	0.43	−0.12	−0.09				
CAT6	15	3	2.46	0.46	0.09	0.00	24 (3)	21 (3)	0.12	0.31
CAT7	15	3	3.31	0.49	0.06	0.05				
SPR1	14	4	3.41	0.45	−0.04	−0.06				
SPR2	14	3	2.97	0.44	0.07	0.01				
SPR3	15	3	2.79	0.44	−0.03	−0.03				
SPR5	14	3	2.79	0.43	0.03	0.05				
SPR6	15	2	2.12	0.49	−0.03	−0.09				
SPR7	15	4	3.78	0.33	−0.11	−0.08				
Mean	14.77	3.00	2.82	0.44	−0.01	−0.01	25.50	22.50	-	0.24
SE	0.44	0.71	0.67	0.01	0.02	0.02	5.74	4.36	-	0.06

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
