# Peer review of "Sociogenetic Organization of the Red Honey Ant (Melophorus bagoti)"

_insects, 2020, doi:10.3390/insects11110755_

Round 1

Reviewer 1 Report

Manuscript ID: insects-974356

Title: Sociogenetic organization of the red honey ant (Melophorus bagoti)

Authors: Nathan Lecocq de Pletincx, Serge Aron

In this manuscript, Lecocq de Pletincx and Aron describe genetic structure of nests and populations in the red honey ant, Melophorus bagoti. The project is quite exhaustive, including description of colony kin structure (queen number, queen mating frequency, male production), basic spatial analysis of population structure and association of worker subcastes (majors and minors)with patrilines. The work is mostly based on DNA microsatellite data on queens, workers and brood, and on top of everything, authors had to first develop their DNA microsatellites by sequencing genomic libraries, identifying DNA microsatellite motifs and designing primers. This is an impressive amount of work for a single article. It is the first study of its kind in this genus and probably also in the tribe Melophorini, increasing the novelty of the work.

The authors show that Melophorus bagoti nests have a single queen which mates with several males, worker male production takes place at least occasionally, populations are random mating and there is weak but significant differentiation between the two study populations, and that the two worker subcastes are not associated with patrilines. As said already, the project is comprehensive, but it seriously suffers from small sample sizes. Small sample size is often unavoidable in social insect research, but it does undermine the generality of many conclusions in this work, which the authors also acknowledge. In any case, the authors should be very careful in their writing. For instance, only two queenright nests produced males and conclusions about the origin of males are tentative at best. Another word of warning goes to leaning too much on the earlier work on breeding behavior of Melophorus bagoti (Schultheiss et al. 2010), this work is also based on a very small number of observations.

I have a few major suggestions, which authors should consider before this paper is published, and a number of minor comments. First, the results are largely based on using worker genotypes to deduce queens’ and their mates’ genotypes. Given that the markers used were nicely polymorphic, the interpretation is probably ok in most cases. However, the confidence of the deduced queen/male genotypes is left hanging in the air. I suggest using Wang’s Colony software to estimate the pedigree relationships to get some confidence for the pedigree. Alternatively, the origin of males could be assessed by using a maximum likelihood methods described in Hastings et al. in Behav Ecol 9:573–.

Table 1. How does equation 1 work to produce larger effective mating frequencies (ke) compared to the observed mating frequency (ka) deduced from worker genotypes of a single mom offspring? ke is supposed be comparable to the effective population size and devalue larger contributions, always resulting in smaller ke (or the same at maximum) than ka. Now ke is larger than ka in 4 nests. An alternative would be to use the within-colony relatednesses to calculate ke, following Ross 1993 (Am Nat, 141: 554-), Queller 1993 (In Keller book) and Seppä 1994 (JEB 7:71-). You actually have many components of the equations available in your results, relatedness among single queen offspring, relatedness between queen and her mates and relatedness among male mates of a single queen.

There is no point in emphasizing the among-colony level differentiation in AMOVA in a system like this, it has no meaning. Each nest represents genetic variation of just one queen and her mates, and among-colony level equals to comparing those individuals to each other. If there is genetic variation in the population, they are bound to be different. In fact, among-colony Fst (=difference among nests) is just a flip side of relatedness (=similarity within nests)

The manuscript is generally well written and fluent, but there are a couple of sections that need revising. First, the presentation of the aims of the study in the end of Introduction is rather weak and not focusing on the main issues of the study. This study is not primarily about estimating inbreeding or genetic differentiation among colonies, no need to emphasize them. Second, the authors have included discussion-type text in the Results section, it can be moved to the Discussion.

Sampling and DNA microsatellite analysis of the eggs, larvae and pupae are described in the Material and Methods section. However, the brood data are hardly used in the manuscript, just the eggs in orphaned colonies. Please explain why you discarded them? There is plenty of interesting information that could be extracted from such data.

Minor issues

Line 119 spell out nSSR, not in common use outside plant kingdom.

Line 158-160 This belongs to the Results

Line 169-170 and later The information about the spatial location of nests is missing, undermining the interpretation of IBD analysis. Isolation by distance cannot be expected if the sampling scale is too short compared to the potential dispersal distance of the sexuals. Please add details of sampling, possibly a map of the sampling sites, and revise the text accordingly.

Line 179-180 and later What does “Queen mating frequency was inferred … using mother-offspring genetic combinations estimated under laboratory conditions” mean? Did you breed them in lab cultures?

Line 180-1, DNA microsatellite analysis of sperm comes up in the data analysis section of Methods, but must be described in the genotyping section first.

Line 227-9 You need to justify why you choose to use loci that were not in HWE. In principal, such loci may not be neutrally behaving Mendelian markers,

Line 274.6 Dissection of workers comes up in the Results, but must be described in the Material and Methods first M

Line 303-4 Please reword: “their genotypes were consistent with the idea that they had been laid by the colonies’ workers”

Line 352 The word Epigenetics is defined as “stably heritable phenotype resulting from changes in a chromosome without alterations in the DNA sequence”, i.e. it involves passing the phenotype to the next generation. I’m not sure if caste determination goes under this label, it takes place within a generation.

Table S1. Add info on which data the frequency of the most common allele, HO and HE are based on.

Author Response

Reviewer 1

Major Revisions

  1. First, the results are largely based on using worker genotypes to deduce queens’ and their mates’ genotypes. Given that the markers used were nicely polymorphic, the interpretation is probably ok in most cases. However, the confidence of the deduced queen/male genotypes is left hanging in the air. I suggest using Wang’s Colony software to estimate the pedigree relationships to get some confidence for the pedigree. Alternatively, the origin of males could be assessed by using a maximum likelihood methods described in Hastings et al. in Behav Ecol 9:573–.

As requested, we now use Wang’s Colony software to confirm pedigree relationships inferred from visual inspection of the genotypes. Note that this method confirmed our estimates by eye.

Lines 166-167.

  1. Table 1. How does equation 1 work to produce larger effective mating frequencies (ke) compared to the observed mating frequency (ka) deduced from worker genotypes of a single mom offspring? ke is supposed be comparable to the effective population size and devalue larger contributions, always resulting in smaller ke (or the same at maximum) than ka. Now ke is larger than ka in 4 nests. An alternative would be to use the within-colony relatednesses to calculate ke, following Ross 1993 (Am Nat, 141: 554-), Queller 1993 (In Keller book) and Seppä 1994 (JEB 7:71-). You actually have many components of the equations available in your results, relatedness among single queen offspring, relatedness between queen and her mates and relatedness among male mates of a single queen.

We have tested other estimators to calculate ke, including those proposed by the Referee. All gave similar results, with ke being sometimes higher than ka. This stems at least in part from that at low sample sizes, several of these estimators tend to overestimate ke values. Yet, the estimator proposed by Nielsen et al. 2003 [ref. 55] performs better than other estimators proposed to date. The reason for ke > ka in some colonies is now explained in the text.

Lines 176-178.

  1. There is no point in emphasizing the among-colony level differentiation in AMOVA in a system like this, it has no meaning. Each nest represents genetic variation of just one queen and her mates, and among-colony level equals to comparing those individuals to each other. If there is genetic variation in the population, they are bound to be different. In fact, among-colony Fst (=difference among nests) is just a flip side of relatedness (=similarity within nests).

We agree with this comment. However, we believe that this information improves the understanding and accessibility of our manuscript, while not increasing its length. First, Fst values are usually given in population genetic studies and we are afraid that discarding this information exposes readers to question about the genetic differentiation among colonies. Second, Fst values are used to compute isolation-by-distance statistics, which are addressed later in the same paragraph. Finally, Fst values are mentioned in two sentences only: one sentence in the Materials and Methods section, and one sentence in the Results section. Therefore, we prefer not to modify the manuscript in accordance with this comment.

  1. The manuscript is generally well written and fluent, but there are a couple of sections that need revising. First, the presentation of the aims of the study in the end of Introduction is rather weak and not focusing on the main issues of the study. This study is not primarily about estimating inbreeding or genetic differentiation among colonies, no need to emphasize them.

As suggested by the Referee, we have re-writen the aims section to make it more focus. Specifically, the sentence on the estimates of inbreeding and genetic differentiation among colonies has been deleted. Moreover, we now highlight that we examined (i) the genetic structure of colonies, (ii) worker morphometry, and (iii) worker reproduction.

Lines 82-87.

  1. Second, the authors have included discussion-type text in the Results section, it can be moved to the Discussion.

As requested, discussion-type sentences in the Results have been moved to the Discussion or deleted from the manuscript to avoid redundance.

Lines 258 and 310.

  1. Sampling and DNA microsatellite analysis of the eggs, larvae and pupae are described in the Material and Methods section. However, the brood data are hardly used in the manuscript, just the eggs in orphaned colonies. Please explain why you discarded them? There is plenty of interesting information that could be extracted from such data.

Brood was found in a small number of colonies and, when present, it occurred in a very limited amount, which prevented relevant studies. To avoid confusion, and because the brood was not used in our study, we do not refer to brood collection anymore.

Lines 95 and 117.

Minor Revisions

- Line 119. Spell out nSSR, not in common use outside plant kingdom.

For simplicity, we changed “nSSR” by “microsatellite”.

Lines 109-111.

- Line 158-160. This belongs to the Results.

As requested, we moved this part of the Methods in the Results section.

Lines 221-224.

- Line 169-170 and later. The information about the spatial location of nests is missing, undermining the interpretation of IBD analysis. Isolation by distance cannot be expected if the sampling scale is too short compared to the potential dispersal distance of the sexuals. Please add details of sampling, possibly a map of the sampling sites, and revise the text accordingly.

To meet the referee requirement, we have added a Supplementary Table (Table S1), with the GPS coordinates of each colony sampled in our study.

- Line 179-180 and later. What does “Queen mating frequency was inferred … using mother-offspring genetic combinations estimated under laboratory conditions” mean? Did you breed them in lab cultures?

The sentence was re-worded as follows: ‘Queen mating frequency was inferred via three methods: (i) using the genotypes of the queens and their worker offspring produced under controlled laboratory conditions ; …’.

Lines 168-169.

- Line 180-1. DNA microsatellite analysis of sperm comes up in the data analysis section of Methods, but must be described in the genotyping section first.

That’s right. The sentence has been moved in section 2.2. Microsatellite markers and genotyping.

Lines 118-119.

- Line 227-9. You need to justify why you choose to use loci that were not in HWE. In principal, such loci may not be neutrally behaving Mendelian markers.

Unpublished data revelead that our results did not change whether we used these markers or not. The point is now clearly stated in the text.

Lines 225-227.

- Line 274.6. Dissection of workers comes up in the Results, but must be described in the Material and Methods.

Indeed. It is now clearly mentioned that we dissected both majors and minors subcastes - see section 2.3.4 Maternity of males of the Materials and Method.

Lines 217-218.

- Line 303-4 Please reword: “their genotypes were consistent with the idea that they had been laid by the colonies’ workers”

The sentence has been reworded as follows: ‘their genotypes were consistent with them being produced by the colonies’ workers’.

Lines 311-312.

- Line 352 The word Epigenetics is defined as “stably heritable phenotype resulting from changes in a chromosome without alterations in the DNA sequence”, i.e. it involves passing the phenotype to the next generation. I’m not sure if caste determination goes under this label, it takes place within a generation.

We have changed ‘epigenetics’ by differential gene expression.

Lines 361 and 366.

- Table S1. Add info on which data the frequency of the most common allele, HO and HE are based on.

Data were based on the genotypes of field workers. This is now clearly mentioned in the caption of Table S2. Please note that Table S1 (first version) is Table S2 in the revised version of our manuscript.

Reviewer 2 Report

The draft entitled “ Sociogenetic organization of the red honey ant ( Melophorus bagoti)” reported that newly developed specific primers for this species and genetic population and colony structure revealed by using these primers. The red honey ant is a famous insect which lives in a harsh environment as an arid zone in Australia, I would be so wondering that there has been no genetical reports until today. The number of primer pairs they developed were 12, which are sufficient numbers to detect variation within colonies and populations. If possible, we need mitochondrial data for this populations, but we will wait for the future works by the authors. The most interesting findings for me is sister- relationship between queen and workers as well as between queen and reproductive as in lines 247-251. These findings suggest that some physical competitions could occur among sisters upon colony inheritance after original queen’s death. In addition, the authors reported a series of novel findings that this species was monogyny and polyandry, and workers reproduced even in queen-right condition irrespective of worker dimorphism: these findings contradicted with a theoretical prediction by the policing theory under genetic background. I think that this finding attracts many readers who are interested in evolution of social insects.

At present, I do not find any fatal flaws on this draft. If the authors will be possible, presenting of queen’s head size would be helpful in Figure 1 (a) as a reference value.

Minor comments: Equation 3, 0,5 -> 0.5 In table 1. Correctly speaking, 0.08 is not mean value instead overall value in the Fisher P value.

Author Response

Reviewer 2

At present, I do not find any fatal flaws on this draft.

Minor Revisions

  1. If the authors will be possible, presenting of queen’s head size would be helpful in Figure 1 (a) as a reference value.

We thank the referee for this relevant comment. As suggested, we have added the maximum head width and the length of the hindleg of 15 queens on Figure 1. We have changed the Materials and Methods and Results sections accordingly.

Lines 188-190; 283-286; Figure 1.

  1. Equation 3, 0,5 -> 0.5

Done.

  1. In table 1. Correctly speaking, 0.08 is not mean value instead overall value in the Fisher P value.

That’s right. The value has been deleted from Table 1; it is now given in the text.

Line 287.
